# Antigen Unmasking Is Required to Clinically Assess Levels and Localisation Patterns of Phospholipase C Zeta in Human Sperm

**DOI:** 10.3390/ph16020198

**Published:** 2023-01-28

**Authors:** Junaid Kashir, Bhavesh V. Mistry, Lujain BuSaleh, Michail Nomikos, Sarah Almuqayyil, Raed Abu-Dawud, Nadya AlYacoub, Hamdan Hamdan, Saad AlHassan, F. Anthony Lai, Abdullah M. Assiri, Serdar Coskun

**Affiliations:** 1Department of Biology, College of Arts and Sciences, Khalifa University, Abu Dhabi 127788, United Arab Emirates; 2Department of Comparative Medicine, King Faisal Specialist Hospital and Research Centre, Riyadh 11564, Saudi Arabia; 3College of Medicine, Alfaisal University, Riyadh 11533, Saudi Arabia; 4College of Medicine, QU Health, Qatar University, Doha 2713, Qatar; 5Department of Pathology and Laboratory Medicine, King Faisal Specialist Hospital and Research Centre, Riyadh 11564, Saudi Arabia; 6Department of Physiology and Immunology, College of Medicine and Health Sciences, Khalifa University, Abu Dhabi 127788, United Arab Emirates; 7Department of Obstetrics and Gynecology, King Faisal Specialist Hospital and Research Centre, Riyadh 11564, Saudi Arabia

**Keywords:** phospholipase C zeta (PLCzeta), oocyte activation, male infertility, sperm, fertilisation, antigen retrieval/unmasking

## Abstract

Mammalian oocyte activation is initiated by intracellular calcium (Ca^2+^) oscillations, driven by the testis-specific phospholipase C zeta (PLCζ). Sperm PLCζ analysis represents a diagnostic measure of sperm fertilisation capacity. The application of antigen unmasking/retrieval (AUM) generally enhanced the visualisation efficacy of PLCζ in mammalian sperm, but differentially affected the PLCζ profiles in sperm from different human males. It is unclear whether AUM affects the diagnosis of PLCζ in human sperm. Herein, we examined whether the application of AUM affected the correlation of PLCζ profiles with sperm parameters and fertilisation capacity. PLCζ fluorescence levels and localisation patterns were examined within the sperm of males undergoing fertility treatment (55 patients aged 29–53) using immunofluorescence in the absence/presence of AUM. The changes in PLCζ profiles following AUM were examined in relation to sperm health and fertilisation outcome. AUM enhanced the observable levels and specific localisation patterns of PLCζ in relation to both optimal sperm parameters and fertilisation outcome, without which significant differences were not observed. The extent of the change in levels and localisation ratios of PLCζ was also affected to a larger degree in terms of the optimal parameters of sperm fertility and fertilisation capacity by AUM. Collectively, AUM was essential to accurately assesses PLCζ in human sperm in both scientific and clinical contexts.

## 1. Introduction

Infertility affects ~15% of couples and afflicts ~7% of men worldwide [1], while ~50% of male infertility is currently unexplained [1,2,3,4]. Although assisted reproductive technology (ART) has alleviated concerns for some conditions, ART outcome seems increasingly determined by the efficacy of oocyte activation, and by association, the profiles of the sperm factor, phospholipase C zeta (PLCζ) [5]. Mammalian oocytes undergo a series of events at fertilisation, termed oocyte activation, driven by unique patterns of intracellular calcium (Ca^2+^) oscillations, and elicited by testis-specific PLCζ, which hydrolyses phosphatidylinositol 4,5-bisphosphate (PIP_2_) to inositol 1,4,5-trisphosphate (IP_3_), in turn acting upon endoplasmic reticulum (ER) IP_3_ receptors to initiate Ca^2+^ release [5,6]. Immuno-depleting PLCζ from sperm extracts reduced oocyte Ca^2+^-release [6], while PLCζ was identified in sperm extract fractions that induced Ca^2+^ oscillations [7,8]. Recombinant PLCζ injection in mouse oocytes resulted in Ca^2+^ oscillations and blastocyst formation [6,9]. RNA interference (RNAi) experiments disrupted testicular PLCζ in mice, with sperm eliciting abnormal Ca^2+^ oscillations and reduced litter sizes [10]. Finally, sperm from PLCζ knock-out mouse models failed to elicit Ca^2+^ oscillations following intracytoplasmic sperm injection (ICSI), exhibiting abnormally high polyspermy and severely reduced patterns of Ca^2+^ release and litter size following in vitro fertilisation (IVF) experiments [11,12].

Sperm from infertile humans unable to activate human and mouse oocytes either failed to elicit Ca^2+^ oscillations completely or did so in a deficient manner (oocyte activation deficiency, OAD) even following ICSI [13]. The PLCζ mutations identified from such patients were predicted to modify the enzyme fold, abrogating sperm PLCζ [14,15,16,17,18,19,20,21]. Reduced/absent PLCζ is indicative of OAD sperm [1,13,16,18,22,23,24,25,26,27,28,29], with PLCζ deficiencies increasingly associated with multiple male-specific conditions [1,16,18,22,23,24,25,26,27,28]. Recently, using two antibodies with demonstrable specificity for PLCζ, Kashir et al. [30] demonstrated that higher levels and specific localisation patterns of PLCζ in human sperm correlated to the optimal ranges of sperm fertility parameters, and higher proportions of successful fertilisation in a general population of males undergoing fertility treatment. Such results indicated that PLCζ potentially represents a diagnostic measure of not just OAD, but also perhaps general male fertility, and may well be influencing the efficacy of subsequent human embryogenesis [31].

Thus, although sperm-PLCζ analysis represents a potentially powerful diagnostic indication of sperm fertilisation capacity [13,14,16,24,31,32,33], numerous concerns have persisted regarding the protocols utilised to examine sperm PLCζ in humans. Indeed, various disparities in antibody specificity and protocol efficiency have yielded conflicting results between studies [15,18,29,32]. The optimal visualisation of PLCζ using highly specific antibodies can only seemingly effectively be performed following the application of antigen retrieval/unmasking (AUM) protocols, while such protocols did not exert a significant change upon the other sperm proteins (such as PAWP) examined apart from PLCζ [32]. It was suggested that perhaps PLCζ could not be optimally visualised without AUM due to the steric/conformational interferences of sperm PLCζ, which prevent antibody/epitope availability [32], potentially explaining some of the disparity present between studies [31,34]. Indeed, it was only after the application of AUM protocols that reliable and repeatable correlations between the levels and specific localisation patterns of PLCζ could be observed, with optimal ranges of sperm fertility parameters and higher proportions of successful fertilisation in a general population of males undergoing fertility treatment [30].

Despite such results, the efficacy of AUM in diagnostic outcomes using PLCζ analysis (levels and localisation patterns) remains a significant question. Indeed, it is not yet clear whether AUM improved or worsened diagnostic prognosis in a larger population [32]. A further detailed analysis is also required following the findings of Meng et al. [35] who reported no significant change following AUM treatment in PLCζ analysis following their in-house protocol. Indeed, numerous lines of opinion concur that a reliable, reproducible, and consistent method is required using specific tools with determined veracity to examine PLCζ in human sperm, without which results cannot be viewed with any reliability [31]

Herein, to examine the efficacy of AUM in PLCζ analysis, we performed for the first time a systematic analysis of PLCζ profiles (levels and localisation patterns) in sperm from males undergoing fertility treatment, both without and with antigen unmasking protocols (no AUM and AUM, respectively). We attempted to examine whether the application of AUM protocols was required for the successful correlation of PLCζ profiles with sperm parameters and fertilisation capacity, and whether AUM differentially affected sperm from different individuals as previously indicated. Such examinations would be essential to ensure that the application of PLCζ as a clinical diagnostic measure is readily and effectively applicable.

## 2. Results

### 2.1. Antigen Unmasking (AUM) Differentially Enhances the Levels of Observable PLCζ in Sperm from Individual Human Patients

Antibody validation experiments revealed that the EF pAb utilized herein recognized a single specific band at the expected size for PLCζ in human sperm (~70 kDa), and also recognised recombinant PLCζ, where both bands were significantly diminished following antibody blocking with purified recombinant human PLCζ protein. Blotting with rabbit IgG did not identify any protein band (Figure 1A–D). Blocking experiments also diminished all patterns of observed PLCζ localisation in the sperm head as well as the tail. No signals were detected using negative controls utilising a secondary antibody only (Figure 1E).

We similarly identified the distributions of PLCζ in the sperm head as previously identified by Kashir et al. [30], as well as in the sperm tail. We identified distinct populations at the equatorial-only (Eq) and acrosomal + equatorial (Ac + Eq) segments, and a ‘dispersed’ pattern of localisation, distributed as punctate loci, throughout the sperm head (Figure 2). A small population of sperm did not exhibit any PLCζ fluorescence in the sperm head.

AUM treatment significantly (*p* ≤ 0.05) increased the total relative fluorescence quantified compared to the total relative fluorescence quantified without AUM (no AUM; 6.3 a.u. vs. 5.7 a.u, respectively; Figure 3A,B). Intriguingly, the level of fluorescence observed in the tail also seemed to increase, although we were unable to further quantify this increase due to the nature of our analytical methods. No significant difference was observed in the variability of PLCζ levels as indicated by the standard deviation of relative fluorescence between the no AUM and AUM groups (2 a.u vs. 1.9 a.u, respectively; Figure 3C).

Two-way ANOVA indicated that the distribution observed between relative fluorescence levels in the no AUM and AUM groups was statistically significant (*p* ≤ 0.05), while all changes in the levels of relative fluorescence quantified before and after AUM for each patient (albeit not in the same cell) were also statistically significant (*p* ≤ 0.05). Finally, two-way ANOVA indicated that status of AUM/no AUM did not determine a change in the relative fluorescence levels, suggesting that this change depended upon the patient examined (but was not related to sperm parameters).

### 2.2. Application of AUM Is Required to Use PLCζ Fluorescence Levels and Localisation Patterns as Indicators of Fertilisation Outcome

To examine whether AUM affected the diagnostic capacity of PLCζ, we examined the potential relationship between the application of AUM and fertilisation outcomes. While the difference in fluorescence levels between unsuccessful and successful fertilisation did not significantly differ in the no AUM group, the analysis of sperm from the same cases following AUM resulted in significantly higher levels of PLCζ fluorescence in cases of successful fertilisation compared to unsuccessful cases (6.3 a.u vs. 3.9 a.u; Figure 4A). Similarly, the Ac + Eq/dispersed localisation pattern ratio did not significantly differ between successful and unsuccessful fertilisation without the application of AUM, while the ratio in the same cases indicated a significantly higher ratio in the cases of successful fertilisation compared to the unsuccessful cases (1.3 a.u vs. 0.9 a.u; Figure 4B). Both distributions were significant as indicated by one-way ANOVA.

### 2.3. AUM Treatment Did Not Affect Observed Proportions of PLCζ Localisation among Sperm Parameters Examined

As previously reported [30], the predominant patterns of PLCζ observed were the Ac + Eq and dispersed patterns of localisation, alongside a smaller population of Eq PLCζ only. Only a minority of sperm exhibited no patterns of PLCζ fluorescence. However, while the distribution of these individual proportions was significant, the application of AUM did not result in a significant change in the proportions observed. The proportions of localisation patterns recorded with or without the use of AUM did not differ either in sperm in general, or in the context of any optimal semen parameters examined, indicating that AUM did not exert any effect upon the overall predominant localisation pattern observed (Appendix A), an observation supported by two-way ANOVA analysis.

### 2.4. Ac + Eq/Dispersed PLCζ Localisation Pattern Ratios Altered Significantly following AUM

To examine whether AUM altered the levels and localisation patterns of PLCζ in relation to the optimal sperm parameters, we performed analyses with and without AUM. While sperm morphology for all patients was examined, we could not perform a satisfactory correlative analysis with PLCζ/AUM as most patients exhibited a high morphology score (88–99%), as low morphology scores were not a specific recruitment criterion for this study. The fluorescence levels of PLCζ did not significantly differ between the non-optimal and optimal ranges of semen volume, sperm motility, and sperm concentration following AUM (Appendix A). However, the application of AUM resulted in a significant drop in the Ac + Eq/dispersed localisation ratio compared to no AUM in relation to the optimal ranges of sperm motility (ratio of 1.5 a.u vs. 2.2 a.u, respectively) and the optimal volume (1.2 au vs. 2.9 a.u, respectively). While the overall distribution was significant, the Ac + Eq/dispersed ratio remained unaltered for the optimal ranges of sperm concentration (Appendix A).

### 2.5. Levels of Change in PLCζ Parameters following AUM Corresponded to Optimal Sperm Parameters

To understand why AUM seemed to exert such a variable effect upon the PLCζ parameters in sperm from different individuals, we examined whether the extent of this change could be correlated with any measure of sperm health or fertility outcome. AUM did not result in a significant change in PLCζ levels and Ac + Eq/dispersed ratios between any parameter examined (Appendix A). However, the same analysis indicated a variable effect with relation to the non-optimal vs. optimal sperm parameters. While the overall distribution observed was significant, the change in PLCζ levels following AUM did not significantly differ for semen volume but was significantly lower in the optimal sperm concentrations compared to the non-optimal sperm concentrations (0.6 a.u vs. 0.8 a.u). However, the opposite was observed for sperm motility, where optimal sperm motility exhibited a higher level of change in PLCζ levels following AUM compared to the non-optimal parameters (0.9 a.u vs. 0.3 a.u; Figure 5A).

While the overall distribution observed was significant, the change in the Ac + Eq/dispersed ratio following AUM was significantly lower in the optimal sperm concentrations compared to the non-optimal sperm concentrations (0.1 a.u vs. 1.3 a.u). However, the opposite was observed for sperm motility and semen volume, where a higher level of change in the Ac + Eq/dispersed ratio following AUM was observed in the optimal parameters compared to the non-optimal parameters (0.6 a.u vs. 0.2 a.u, and 1.6 a.u vs. 0.2 a.u, respectively; Figure 5B). The examination of the change in fluorescence levels and Ac + Eq/dispersed ratio in relation to fertilisation outcomes revealed that the change in relative fluorescence levels was significantly higher in the cases of successful fertilisation compared to the unsuccessful cases (0.8 a.u vs. 0.2 a.u, respectively; Figure 5C), while there was no significant difference in the Ac + Eq/dispersed localisation ratio between the successful and unsuccessful fertilisation groups (0.11 a.u vs. 0.12 a.u, respectively; Figure 5D).

## 3. Discussion

Ca^2+^ profiles induced by abrogated PLCζ are increasingly associated with certain conditions of male infertility, or even perhaps subfertility, arising from defective oocyte activation. Sperm from infertile men consistently failed to activate oocytes and could either not elicit a Ca^2+^ release in mouse oocytes or did so abnormally [13,16]. Such sperm exhibit abrogated PLCζ [1,13,14,15,16,17,18,19,20,22,23,25,26,27,28,36,37,38,39,40], suggesting that PLCζ defects underlie, or at least contribute to, such cases of fertilisation failure. Clinically, complete fertilisation failure is attributed to sperm-specific defective oocyte activation, rather than any other cause [1,31]. The most rapidly applicable approach is the immunocytological analysis of sperm PLCζ [1,13,14,15,16,17,18,19,20,22,23,25,26,27,28,36,37,38,39,40]. However, varying protocol efficiencies have yielded conflicting outcomes between studies examining human sperm PLCζ [15,18,29,32]. The optimal visualisation of PLCζ using specific antibodies could only effectively be performed following the application of antigen retrieval/unmasking (AUM) protocols [32], which most sperm PLCζ studies have not incorporated. Indeed, AUM protocols do seem to be benefitting other areas of sperm-based research, being recently utilized to examine Na^+^/H^+^ exchangers (NHEs) in the plasma membrane of porcine sperm [41]. Kashir et al. [32] also suggested that AUM exerted a differential effect in sperm from different human male samples [32], indicating a potentially variable effect of AUM upon individual patients.

Meng et al. [35] recently suggested that AUM was not required for optimal PLCζ visualisation, and that a routine method is sufficient, if not superior, to AUM protocols. Indeed, the authors suggested that such protocols (specifically acidic methods of AUM) were counterproductive rather than beneficial. Another aspect suggested by Meng et al. [35] was that antibody specificity and variability may also determine the efficacy of AUM as determined by epitope specificity. However, several of the arguments made by Meng et al. [35] remain flawed, specifically since the antibody used by these authors seems notoriously unspecific when examined for antigen binding using immunoblotting, identifying multiple bands. Indeed, the authors did not perform such confirmations of specificity in their analyses unlike in our experimentation. Indeed, while peptide blocking was used by the authors to demonstrate immunofluorescence specificity, this can only be reliable if the protein of interest is specifically targeted by the antibody in question (as determined by immunoblotting). This contrasts with immunoblotting using our antibody, which has repeatedly been demonstrated to identify specifically single bands corresponding to PLCζ, with immunofluorescent specificity also determined by epitope blocking.

Another issue in the study by Meng et al. [35] is that a like-for-like analysis was not performed, whereby the authors’ in-house protocol was not subject to AUM protocols. The protocols used by Kashir et al. [32] differed in numerous ways from that of the in-house protocol deployed by Meng et al. [35] in terms of antibody concentration and blocking conditions, among others. Thus, since the same conditions were not applied when examining the in-house protocols, the purported like-for-like analysis required was not necessarily performed. Finally, the authors examined a relatively smaller number of patients in their study, far fewer than our current efforts. Hence, such findings at the very least merit further investigation to ascertain whether AUM is indeed beneficial for clinical application, particularly since Meng et al. [35] raise questions regarding the applicability of AUM.

Aras-Tosun et al. [42] could not find any correlations between PLCζ and fertilisation, and this is another example of a study that did not employ AUM in their investigative protocols. Intriguingly, however, the authors reported fertilisation in cases despite low levels of sperm PLCζ, suggesting that perhaps not all the PLCζ was being detected. The experiments could therefore have benefitted from AUM application, as indicated in our results, i.e., it was only after AUM that correlations could be observed. To this end, it is necessary to expand the veracity of the clinical applicability of AUM towards sperm PLCζ analysis. Herein, we examined in human sperm from a general population of males undergoing fertility treatment (in the absence and presence of AUM; no AUM and AUM, respectively) whether the application of AUM protocols was required for the successful correlation of PLCζ profiles with sperm parameters and fertilisation capacity, and whether AUM differentially affected sperm from different individuals as previously indicated.

### 3.1. AUM Is Required to Reveal Significantly Higher PLCζ Parameters in Cases of Successful Fertilisation

Kashir et al. [32] demonstrated that AUM was required to enhance the visualisation efficacy of PLCζ in mammalian sperm but affected sperm from different patients differently. Herein, we made the same observations following our experiments. Our investigations further indicated that AUM did not reduce the levels of variability, suggesting that PLCζ variability may be due to the levels of the enzyme itself, as opposed to the accessibility of the antibody to its epitope [32]. In concordance with recent findings [30], we identified four major patterns of PLCζ localisation at the equatorial only (Eq), acrosomal + equatorial (Ac + Eq) segments of the sperm head, and a dispersed pattern of punctate loci throughout the sperm head. We also identified a minority of sperm where PLCζ fluorescence was not detectable. It should be noted that these localisation populations were demonstrably PLCζ-specific through a combination of antibody blocking and omission experiments using the same polyclonal antibody we currently utilise (EF pAb) [30]. To this degree, the various localisation patterns observed can be attributed to PLCζ, considering the blocking or exclusion of the antibody used diminished all patterns of localisation, while immunoblotting further confirmed the specific binding of this antibody to sperm PLCζ.

Kashir et al. [30] indicated that levels of PLCζ were significantly higher in sperm leading to successful fertilisation compared to unsuccessful fertilisation, further suggesting that the Ac + Eq localisation pattern was predominantly present in sperm with better quality parameters (motility, concentration, volume), while dispersed PLCζ was more predominant in sperm with lower parameters of sperm health. Furthermore, the ratio between these two patterns (Ac + Eq/dispersed ratio) was significantly higher in cases of successful fertilisation compared to cases of fertilisation failure. In line with these findings, we observed that AUM significantly increased the levels and the Ac + Eq/dispersed ratios of PLCζ. It was only after application of AUM that the levels of these parameters were significantly higher in the successful fertilisation group compared to the unsuccessful fertilisation group, suggesting that AUM application is required to more accurately assess potential diagnostic outcomes using PLCζ immunological analysis, particularly since AUM enhances the visualisation efficacy of PLCζ [30,31,32,34]. It is also interesting that AUM enhanced the PLCζ fluorescence observed throughout the sperm, including the fluorescence observed in the tail. However, while we have previously asserted that perhaps the tail fluorescence observed indeed represents a population of PLCζ [30], the functional significance of this remains to be investigated. Furthermore, given the nature of our analyses, we were unable to quantify the increase in tail fluorescence, and thus cannot comment on the extent of this change. It is thus imperative that future studies attempt to take into account the potential tail population of PLCζ.

### 3.2. AUM Altered the Ac + Eq/Dispersed PLCζ Ratio in Relation to Optimal Sperm Parameters

Similar to previous observations [30], the predominant patterns observed throughout all our analyses were the Ac+Eq and dispersed patterns of PLCζ. However, AUM did not seem to exert a significant change on the proportions of localisations observed, either in sperm in general, or in relation to the optimal parameters of sperm health. Furthermore, a similar observation was made for total levels of PLCζ, whereby the application of AUM did not result in a significant change in levels in relation to the optimal sperm parameters, indicating that although AUM was found to enhance the total levels of PLCζ visualisation efficacy, this alteration was independent of the individual sperm parameters. However, a significant decrease in the Ac + Eq/dispersed PLCζ ratio was observed following AUM in the optimal ranges of sperm motility and volume, but not of concentration, suggesting that AUM increased the proportion of observable dispersed PLCζ in relation to sperm motility and volume (but not concentration), thereby decreasing the Ac + Eq/dispersed ratio.

Increasing sperm quality corresponded to decreasing proportions of sperm with dispersed localisation, while conversely corresponding to elevated proportions of sperm with Ac + Eq PLCζ [30]. As sperm quality improved, the proportion of sperm with Ac + Eq PLCζ increased, while dispersed PLCζ decreased. This suggested that Ac + Eq PLCζ is most physiologically relevant, while the dispersed localisation patterns corresponded to at least sperm with a decreased oocyte activation and fertilisation capability. Dispersed/punctate PLCζ have been associated with abnormal sperm parameters such as motility and/or morphology, and a reduced capacity for oocyte activation and fertility outcomes [13,16,18,29,43,44]. PLCζ localisation was associated with total levels of PLCζ, with the equatorial and dispersed patterns correlating to lower levels, and the Ac + Eq pattern correlating to higher levels of PLCζ, indicating that a higher Ac + Eq/dispersed PLCζ ratio correlated to better quality sperm, while a lower ratio corresponded to sub-optimal levels [30]. To this degree, our results indicated that such an examination without the application of AUM could result in artificially higher ratios, potentially resulting in skewed diagnoses of sperm health.

Puzzlingly, AUM did not significantly alter the levels in relation to any sperm parameter examined. However, this finding is in line with assertions that levels of PLCζ are indicative of fertilisation success rather than sperm health [30,31,34], as levels of PLCζ did not exhibit any significant relationship with any sperm parameter examined. Similarly, we observed that levels of PLCζ were closely linked to fertilisation success, while specific localisation patterns corresponded to the parameters of sperm health. To this degree, our results further indicate that AUM is a requirement to accurately assess both of these parameters.

### 3.3. The Level of Change in PLCζ Parameters following AUM Is Related to Optimal Sperm Parameters and Fertilisation Success

No significant correlations could be observed between the extent of change in levels and the localisation ratio of PLCζ and optimal/non-optimal sperm parameters. However, this was perhaps attributable to the relatively low patient population (55 males) that was examined in this study which may prevent robust conclusions being made, despite significant differences already being observed. It is thus essential for similar studies on a larger scale to validate our findings through a multi-centre analysis of a significantly larger population of patients before any conclusive correlations (or lack thereof) can be determined.

However, we report significant differences in the extent of change in levels and localisation ratios of PLCζ following AUM in relation to optimal sperm parameters. At the ideal levels of sperm motility, AUM induced a significantly larger change in both levels and localisation ratios of PLCζ compared to sperm exhibiting non-optimal motility. Meanwhile the optimal ranges of sperm concentration exhibited the opposite for both localisation ratios and levels of PLCζ. For the optimal parameters of sperm volume only the localisation ratios of PLCζ exhibited a significant difference in optimal levels compared to the non-optimal parameters, while no significant difference was observed for fluorescence levels. While it is clear that AUM exerts a differential effect, and perhaps this accounts for the differential individualistic effect of AUM upon sperm from different males, the reason(s) underlying the opposing effects of AUM for optimal sperm motility and sperm concentration remain unclear. Perhaps such questions could be answered following the further elucidation of why AUM seems to be required to reveal the optimal patterns and visualisation efficacy of PLCζ in mammalian sperm.

Furthermore, AUM evidently induced a significantly larger level of change in cases of successful fertilisation compared to unsuccessful cases. However, this was only observed with the levels of PLCζ fluorescence, and not the Ac + Eq/dispersed localisation ratio. This is perhaps further in line with our and previous assertions that levels of PLCζ are more directly indicative of sperm fertilisation capacity, while localisation patterns perhaps better indicate sperm health in terms of motility, concentration, and volume [30]. Another explanation could be interactions either with PLCζ oligomers, or with other modulatory protein(s) in tight interactions with PLCζ, based on the thinking that such strong interactions would prevent antibody access to the epitope [32].

AUM enhanced the observable levels of PLCζ in sperm, while also increasing the proportion of sperm exhibiting a dispersed pattern of PLCζ localisation (as indicated by the Ac + Eq/dispersed localisation ratio). This perhaps indicates that the reason AUM seemingly enhances PLCζ visualisation efficacy and is required to reveal significant PLCζ diagnostic parameters in relation to successful fertilisation, is that AUM reveals a population of previously undetectable dispersed PLCζ in the sperm. Indeed, recent findings suggested that a dispersed pattern of PLCζ corresponded to low fluorescence levels of PLCζ in the sperm head and a reduced capacity for oocyte activation and fertility outcomes [13,16,18,29,30,31,32,43,44]. Without the application of AUM, studies have previously indicated that higher proportions of sperm exhibiting a complete absence of PLCζ in the sperm head could be linked to ICSI failure [18,29,33,45]. Perhaps such cases where the sperm did not exhibit detectable levels of PLCζ in fact represent dispersed PLCζ which requires AUM to become detectable [32]. Indeed, higher proportions of dispersed PLCζ corresponded to lower proportions of sperm with absent/low sperm-PLCζ levels and poorer sperm quality [30]. Our current results also supported assertions such as that the Ac + Eq/dispersed ratio of PLCζ localisation decreased following AUM in optimal sperm parameters, indicating an increase in the proportion of sperm exhibiting dispersed PLCζ following AUM.

Perhaps an avenue of investigation required would be that proposed by Meng et al. [35], who suggested that AUM success would be dependent upon the antibody/epitope utilised, which is of course possible given the local biophysical properties influencing epitope accessibility within protein structures. However, this would require a focused study, using antibodies of demonstrable specificity in both immunoblotting and immunofluorescent analyses. It is possible that another aspect which would further influence results could be the specific method of AUM utilised. Indeed, there are numerous methods of AUM that have been utilised in the literature, both mechanical and enzymatic, both of which were examined by Kashir et al. [32] in the context of PLCζ in mammalian sperm. In fact, this study indicated that perhaps different methods of AUM were more suited for the PLCζ of various species, with heat-induced AUM more suited for mouse PLCζ, while the chemically based acid tyrodes treatment method seemed to better suit human sperm. This is perhaps attributable to the varying solubility between human and mouse PLCζ, where mouse PLCζ is thought to be considerably more insoluble than human PLCζ in sperm [1,5], and thus perhaps requires more of an invasive protocol [32]. Therefore, while we relied on the results of Kashir et al., [32] who indicated that the acid tyrodes method of AUM was most effective (and convenient) for human sperm, it is possible that perhaps other methods of AUM may yet yield improved results and should be examined by subsequent studies.

Another aspect worth considering is the phenomenon of inter- and intra-sperm variability observed by this and previous studies in the levels and localisation patterns of PLCζ, most prolifically by Kashir et al. [18] who found that such was the issue of PLCζ variability that this could perhaps limit the applicability of PLCζ as a diagnostic measure if not examined correctly. However, variability is not a PLCζ-specific phenomenon in sperm (particularly in humans), with variability noted for all aspects of sperm parameters [46,47], which may perhaps be seasonal in nature [48]. Only following large-scale, multi-centre examinations was an internationally accepted range of parameter conditions established for such sperm parameters. A similar set of requirements is necessary for PLCζ as well, where many conditions and individuals are evaluated before an acceptable range of parameters associated with clinical outcomes may be established. However, before this can be carried out, it is necessary that the correct methods of examination are applied, otherwise any associations observed stand the risk of being artificially skewed or non-representative, especially for a factor such as PLCζ which has yet to reveal its mechanisms of regulation within sperm, including any regulating/interacting partners. To this end, we hope that our current manuscript would aid in establishing this optimal set of conditions required to establish a relevant range of PLCζ parameters and associations, to be utilised in a clinical context.

## 4. Conclusions

In the present study, we report that AUM allowed PLCζ to be related to both optimal sperm parameters and fertilisation outcome, without which significant differences were not observed. Furthermore, the extent of the change in the levels and localisation ratios of PLCζ was also affected to a larger degree in terms of the optimal parameters of sperm fertility and fertilisation capacity by AUM. The application of AUM seems to be required to provide a more accurate analysis of PLCζ in mammalian sperm in both scientific and clinical contexts. While it is true that the relative size of our examined population was relatively low, the number and extent of cases analysed were among the higher end compared to similar studies in the field, and were adequate to reveal significant differences. The observable levels of PLCζ exhibited a much larger degree of change in successful rather than unsuccessful fertilisation groups (between AUM vs. no AUM). Previously, we have asserted that AUM is required to fully visualise PLCζ due to its associations with either PLCζ or other interactors. In this way, our current data, while indeed preliminary, may suggest that this interaction is strongest or more prevalent in the successful fertilisation group. Herein, we show that AUM is required to accurately utilize PLCζ as a clinical indicator of sperm fertility (at least in relation to optimal sperm parameters and successful fertilisation rates). These findings require further replication and translation in a larger multicentre trial to hasten the diagnostic promise of PLCζ in the clinic, not only for cases of OAD, but also potentially more general cases of male infertility and/or sub-fertility.

## 5. Materials and Methods

### 5.1. Patient Recruitment and Semen/Sperm Processing

The participants in this study were couples undergoing consecutive cycles of assisted reproductive technology (ART) treatment at the King Faisal Hospital and Research Centre (KFSHRC) IVF clinic in Riyadh, Kingdom of Saudi Arabia between 2018 and 2021. The KFSHRC local research ethics committee and the office of research affairs approved the study (RAC# 2170015, first approved in 2017 and renewed yearly thereafter), and all enrolled patients gave their written informed consent before the treatment and experimentation. As we aimed to ascertain the correlations between phospholipase C zeta (PLCζ) and general sperm parameters, our sole recruitment criterion for males was that they had to exhibit a minimum sperm concentration of 5 × 10^6^ sperm/mL, while all female patients had to have at least 5 oocytes (i.e., couples would definitely be undergoing treatment).

Fifty-five patients were recruited (29–53 years old), and semen samples were obtained onsite via masturbation within a sterile plastic container after 2–7 days of abstinence. The semen was liquified up to 60 min prior to analysis. During the treatment preparation, semen analyses, including sperm volume, concentration, and motility, were performed and evaluated following the standard WHO recommended protocols [49] (Appendix A), at the KFSH&RC Assisted Reproductive Technology Laboratory, which is accredited to perform semen analysis by the College of American Pathologist (CAP). Semen analysis quality control was achieved by the daily checking of pre-recorded videos and running Accu-Beads (Hamilton Thorne, Inc., Beverly, MA USA). The laboratory also participates in CAP’s semen analysis external proficiency testing. Fertilisation success was determined by observing second polar body extrusion and the formation of two pronuclei. Further semen and sperm processing, and density gradient washing (DGW) were performed as previously described [30].

For fixation, the sperm suspensions were centrifuged at 500× *g* for 10 min, followed by resuspension in 4% paraformaldehyde and incubation at RT for 15 min. Following fixation, the sperm were centrifuged at 500× *g* for 10 min, and the pellet was washed thrice using PBS + protease inhibitors and resuspended in a solution appropriate for the size of the pellet. For the immunoblotting procedures, following the determination of sperm concentration, the appropriate volumes of sperm to give 500,000 sperm/aliquot were added to a 5× sample loading (Laemmlli) buffer (10% (*w*/*v*) SDS; 10mM beta-mercaptoethanol; 20% (*v*/*v*) Glycerol; 0.2 M Tris-HCL, PH 6.8; 0.05% (*w*/*v*) bromophenol blue). Each aliquot was briefly vortexed, snap frozen in liquid nitrogen, and stored at −80 °C until required [30].

### 5.2. Immunofluorescence Microscopy

The fixed sperm were added to slides previously coated with 0.01% (*w*/*v*) poly-l-lysine solution (Sigma Aldrich, Gillingham, UK), within hydrophobic moulds drawn using a PAP pen (Vector laboratories). Sperm permeabilisation was performed with PBS-1% Triton X-100 (*v*/*v*) for 1 h at RT and blocked by a PBS-10% bovine serum albumin (BSA; Sigma Aldrich, UK). The primary EF pAb (diluted 1:50 with PBS-5% BSA) was added overnight at 4 °C, following which the AlexaFluor-488 conjugated goat anti-rabbit secondary antibody (1:100; Life Technologies, Paisley, UK; catalogue number: A48282), diluted in PBS-5% BSA was added for 1 h at RT. Washing with PBS was performed between all steps. The samples treated with secondary antibody dilutions only were used as negative controls for the immunofluorescence imaging. The slides were mounted using the Vectashield mounting medium containing 4′-6-diamidino-2-phenylindole (DAPI) (Vector Laboratories). Immunofluorescence was performed in the presence/absence of antigen unmasking/retrieval (no AUM/AUM, respectively) using acidic Tyrode’s solution (AT) (pH = 2.5–3.0; Sigma, Welwyn Garden City, UK), included in the immunofluorescence methodology following permeabilisation and before blocking. The images were captured at the same exposure for all patients and samples. The extensively characterised EF polyclonal antibody (EF pAb; raised in rabbits against a 16-mer-human PLCζ peptide sequence—8SKIQDDFRGGKINLEK23) was used in this study [30,32,36,50,51,52].

The acquisition of images was performed as previously described by Kashir, et al. [30]. by capturing images at ×40 and ×100 magnifications (using oil immersion, type FF, Electron Microscopy Sciences, catalogue number: 16916-04), using an OLYMPUS BX53 fluorescence microscope (Olympus, Breiningsville, PA, USA). An OLYMPUS DP73 camera (Olympus, Breiningsville, PA, USA) was used to capture the images using OLYMPUS cellSens entry software v1.17 (Olympus, Breiningsville, PA, USA). The brightfield images were captured alongside the corresponding fluorescence images obtained using a fluorescein isothiocyanate (FITC) filer. All images were captured at the same exposure time throughout patients and samples [30].

### 5.3. Immunoblotting and Antibody Validation

The single-use sperm aliquots were thawed and heated at 101 °C for 5 min, vortexed, cooled on ice for 5 min, and briefly centrifuged before being loaded onto gels. The sperm protein samples (500,000 sperm/lane) were separated through SDS-PAGE using 10% gels and transferred onto polyvinylidene difluoride (PVDF) membranes (Amersham Hybond, GE Healthcare Life Sciences, Piscataway, NJ, USA), using wet-transfer at 100 V for 1 h. The transfer and protein separation efficacy was determined by staining the membranes with 0.1% (*w*/*v*) ponceau stain (in 5% (*v*/*v*) acetic acid). The membranes were incubated overnight at 4 °C with the primary EF pAb (diluted 1:1000), followed by incubation with an anti-rabbit secondary antibody conjugated with horseradish peroxidase (HRP) for 1 h at RT, diluted at 1:1000. HRP detection was achieved using the ECL select kit, following the manufacturer’s recommended protocol (GE Life Sciences, Belfast, UK). Chemiluminescence was detected using the ImageQuant LAS4000 (GE Healthcare Life Sciences, USA) image system [30].

Antibody specificity was determined through blocking experiments [30], using a molar excess of affinity-purified recombinant NusA-PLCζ protein before addition to the membranes for immunoblotting. The recombinant NusA-tagged PLCζ bacterial cell lysates were prepared, and the recombinant NusA-tagged PLCζ was purified [36,50,51,52]. Briefly, a 10-fold molar excess of recombinant protein and a comparative antibody (10 M recombinant protein to 1 M antibody) were co-incubated in respective antibody incubation solutions for immunoblotting and immunofluorescence experiments for 1 h at RT, alongside a separate incubation with only the primary antibody in antibody incubation solution. Both incubations were then supplemented with the required volumes of 5% BSA-PBS to reach the required dilutions, which were then added to the membranes/slides for overnight incubation. In all the immunofluorescence and immunoblotting methods, controls were also conducted by incubating secondary antibodies only in the absence of primary antibodies for primary antibody incubation, as well as with rabbit IgG only (ThermoFisherScientific, Horsham, UK; Catalog: 02-6102) [30].

### 5.4. Sperm PLCζ Analysis

The sperm PLCζ fluorescence quantification and statistical analysis was performed on images obtained at 40× magnification as previously described [18,30,32] using the ImageJ software package (National Institutes of Health, Bethesda, MD, USA) with the regions of interest tool [18,30,32]. The analysis was performed for both without and with AUM samples. A localisation pattern analysis also examined the ratios between the acrosomal + equatorial/dispersed localisation patterns [30]. As the number of slides examined would differ between patients due to sperm concentration, the main measure of analysis was set at 100 cells/patient (no AUM/AUM) to be examined, and the total fluorescence subtracted from background fluorescence to yield the relative fluorescence. The change in relative fluorescence and the Ac + Eq/dispersed localisation ratio following AUM was also recorded by subtracting the values obtained without AUM from those observed with AUM, to quantify the effect of AUM. The statistical analyses were performed using Prism 7.0 (Graphpad, San Diego, CA, USA) [18,30,32,53]. A *p*-value ≤ 0.05 was considered statistically significant. All the image capturing and analyses were performed by the same researcher to remain consistent in the analyses, with independent concurrent verification by a separate researcher.

The differences between the two variables were examined using the *t*-test with Welch’s correction for unequal standard deviations. Multi-variable examinations were performed (including multiple comparisons tests) using either one-way or two-way analysis of variance (ANOVA), followed by Tukey post-hoc analysis for one-way ANOVA, and the two-stage step-up method of Benjamini, Krieger and Yekutieli [53] for two-way ANOVA, in order to control the false discovery rate following multiple comparisons. Data represented as proportions (%) were ARCSIN transformed prior to the statistical analysis to avoid the truncation of data. The significance levels were adjusted for localisation pattern correlations, where the post-hoc corrected significance value was adjusted according to the number of localisation patterns observed (0.05/n, where n is the number of distinct patterns observed). For the localisation pattern analyses a *p*-value ≤ 0.0125 was considered significant.

## Figures and Tables

**Figure 1 pharmaceuticals-16-00198-f001:**
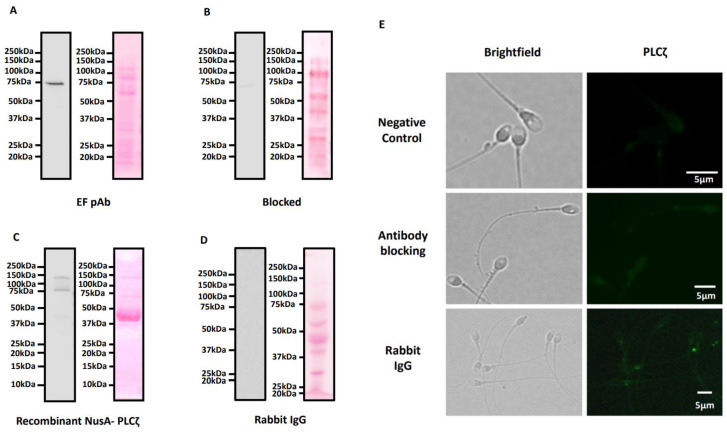
Immunoblotting analysis and corresponding ponceau-stained images indicating the specificity of the EF polyclonal antibody in recognising PLCζ in human sperm. The EF pAb recognised a single band at the expected molecular weight for human sperm and recombinant PLCζ (~70 kDa) (**A**,**C**). This band was depleted following blocking with recombinant NusA-PLCζ protein (~150 kDa) (**B**). Immunoblotting with rabbit IgG did not identify any protein bands (**D**). (**E**) Representative images of secondary antibody incubation only, without a primary antibody: top panel, negative control, and antibodies blocked with a molar excess of purified recombinant NusA-PLCζ; middle panel, antibody blocking, and sperm treated with rabbit IgG as primary antibody treatment; bottom panel, rabbit IgG. Brightfield (left panels) and PLCζ (green fluorescence, left panels) images were obtained. Both panels indicated diminished/absent levels of PLCζ fluorescence. Representative images obtained were captured at 100× and white scale bars represent 5 μm. Images are representative of 100 cells examined for each treatment.

**Figure 2 pharmaceuticals-16-00198-f002:**
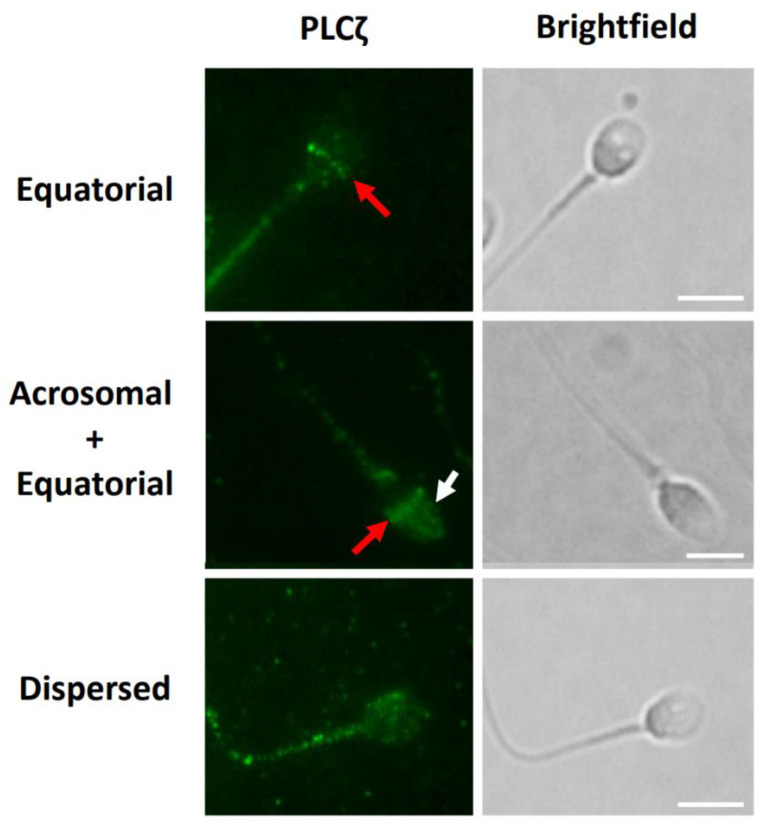
Representative observations of PLCζ localisation patterns in human sperm under both AUM and no AUM conditions, indicated as appropriate. Representative equatorial (red arrows), acrosomal + equatorial (white arrow indicates acrosomal localisation), and dispersed patterns of localisation are indicated. Fluorescence was also observed throughout the tail and midpiece of all sperm examined. Images obtained were captured at 100× and white scale bars represent 10 μm. Images are representative of 100 cells examined per patient recruited (55 patients), for both no AUM and AUM analyses.

**Figure 3 pharmaceuticals-16-00198-f003:**
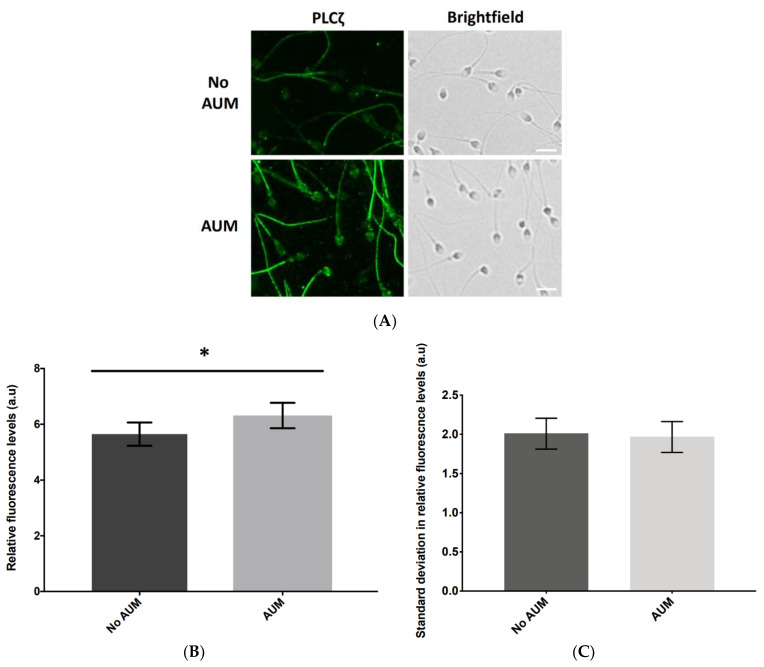
(**A**) Representative immunofluorescence images of observed PLCζ relative fluorescence in human sperm without and with antigen unmasking (no AUM and AUM, respectively), indicating (**B**) the increase in observable fluorescence following AUM which was significantly higher than the fluorescence observed without AUM. (**C**) Histogram indicating the change in the average variation of relative fluorescence levels of PLCζ (measured by average standard deviation, SD) with (light bars) and without (dark bars) antigen unmasking (AUM and no AUM, respectively). Asterisks (*) indicate a statistically significant (*p* ≤ 0.05) difference. Data are indicative of 100 cells examined from three technical repeats of sperm each obtained from 55 patients, for both no AUM and AUM treatments. Images were captured at 40× magnification and white scale bars represent 10 μm.

**Figure 4 pharmaceuticals-16-00198-f004:**
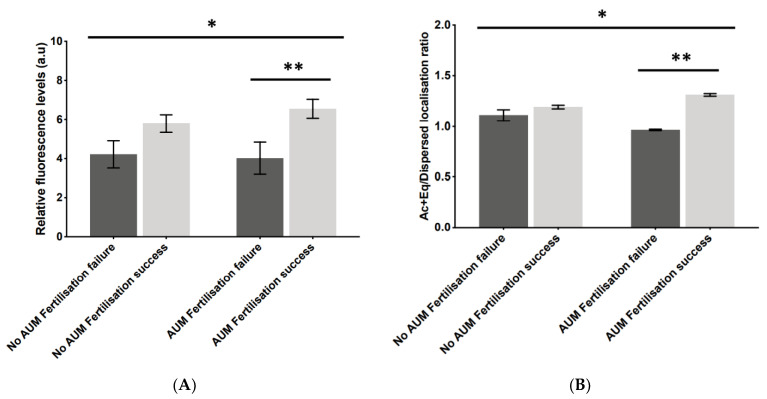
Histograms indicating the differences in (**A**) relative fluorescence levels and (**B**) acrosomal + equatorial/dispersed localisation pattern ratios of PLCζ in cases of fertilisation failure (dark bars) compared to cases of fertilisation success (light bars), in the same cases without (left-hand bars) and with (right-hand bars) antigen unmasking (no AUM and AUM, respectively). Asterisks (*, **) indicate a statistically significant (*p* ≤ 0.05) difference. Data are indicative of 100 cells examined from 55 patients each for both AUM and no AUM groups. a.u.: arbitrary units.

**Figure 5 pharmaceuticals-16-00198-f005:**
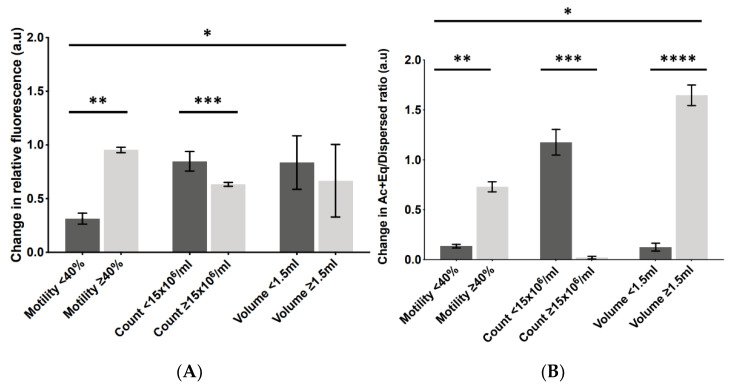
Representative indication of the change in (**A**) PLCζ fluorescence levels and (**B**) Ac + Eq/dispersed localisation ratios following AUM in non-optimal (dark bars) versus optimal (light bars) parameters of sperm motility (≥40%), sperm concentration (count; ≥15 × 10^6^ sperm/mL), and semen volume (1.5–5.5 mL). (**C**) PLCζ fluorescence levels and (**D**) Ac + Eq/dispersed localisation ratios following AUM in cases of unsuccessful fertilisation (dark bars) compared to cases of successful fertilisation (light bars). Asterisks (*, **, ***, ****) indicate a statistically significant (*p* ≤ 0.05) difference. Data are indicative of 100 cells examined from 55 patients each (550 cells). a.u.: arbitrary units.

## Data Availability

The datasets generated and/or analysed during the current study are not publicly available as this includes anonymised patient data which cannot be made publicly available as per the requirements of the ethical permission granted for this study.

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
