# Peer review of "Antigen Unmasking Is Required to Clinically Assess Levels and Localisation Patterns of Phospholipase C Zeta in Human Sperm"

_pharmaceuticals, 2023, doi:10.3390/ph16020198_

Round 1

Reviewer 1 Report

General comment

The manuscript entitled “Antigen unmasking is required to clinically assess levels and localization patterns of phospholipase C zeta in human sperm” aims to examine the efficacy of AUM in PLCζ analysis in males undergoing ART with and without AUM protocol. The study is well-written, and comprehensive and deals with an interesting and currently ongoing topic. Overall few corrections would be required to improve the quality of the work. These corrections have to be considered more as suggestions to improve the study's readability, which is indeed well done.

INTRODUCTION

I would place the section regarding the epidemiology of infertility before introducing the PLCζ.

“thus, while sperm PLCζ analyses…” – check grammar and sentence structure.

“However, while Kashir et al.” – This section could be shortened and reported in the discussion.

RESULTS

Albeit the results are well described, the description of the figures seems in a few cases too intrusive and redundant with the text reported in the paragraph. I would suggest shortening the description of the figures and moving that information in the text when needed.

DISCUSSION

As for the results, the discussion is thoroughly reported. Nevertheless, it would probably be less noisy if another paragraph regarding the limitations would be added.

CONCLUSION

This section should be shortened and summarize the conclusions and the future perspectives related to your study.

MATERIALS AND METHODS

Try to synthesize when possible

too many times "as previously described" is reported. I know that the procedures have been already described elsewhere but I think minimal explanations could benefit the work. Or at least try to avoid the redundancy of the expression.

Reviewer 2 Report

The authors provide useful in clinical laboratories where fertility parameters are studied and information on protocols use for phospholipase C zeta fluorosteining in spermatozoa. The manuscript is described clearly including tiny details. However other information should be provided before acceptance.

1. Were the spermatozoa checked first for concentration, motility, and morphology? Provide mean values of these analyses.
2. Provide full information on used all chemicals (supplier, number etc.) and concentration of used antibodies.
3. How many slides were examined for one patient, who examined it?
